**Data Availability Statement:** The data cannot be published, since the exclusive rights to the database belong to the Far Eastern Scientific Center of Physiology and Pathology of Respiration

# Differences in the health-related quality of life in patients with asthma living in urban and rural areas in the Amur Region of Russian Federation

**Natalia L. Perelman**👤*◉, **Victor P. Kolosov**◉

Laboratory of Prophylaxis of Nonspecific Pulmonary Diseases, Far Eastern Scientific Center of Physiology and Pathology of Respiration, Blagoveshchensk, Russian Federation

◉ These authors contributed equally to this work.
* lvovna63@bk.ru

## Abstract

### Background

Asthma usually arises from an interaction between host and environmental factors. Growing attention has been paid to a place of residence as a factor shaping health-related quality of life (QoL). This study investigated the rural-urban disparity in QoL among adult asthma patients in the Amur region of Russian Federation.

### Materials and methods

This cross-sectional study included 351 randomly selected adults with asthma. We analyzed QoL (SF-36 and AQLQ scores), asthma control (ACT), and anxiety and depression (HADS) depending on the place of residence (urban *vs.* rural).

### Results

The scale "Role Emotional" (RE) of SF-36 was significantly lower in patients from rural areas compared to urban residents (59.3±3.1 vs. 70.4±2.3 points; p = 0.0042). In the urban group, the correlation analysis demonstrated a clear influence of RE on patients' own assessment of their physical functioning (PF, r = 0.53; p<0.0001). Both groups demonstrated low "Social Functioning" (SF). In the group of urban residents, correlation analysis revealed the presence of positive correlations of SF-36 domains reflecting physical (PF, RP, BP) and social functioning (SF, VT) with most scales of both QoL questionnaires. The domains of the emotional sphere (RE and MH) positively correlated with all scales of both QoL questionnaires among urban residents. In the group of rural residents, a comparative analysis showed the absence of significant correlations between more of the QoL scales. Although Asthma Control Test did not differ between groups, we noted a significantly higher need for $\beta_2$-agonists in rural areas compared to urban areas (4.2±0.6 vs. 2.7±0.3 inh/day, respectively; p = 0.0221). The frequency of urban residents with a clinically significant level

(Certificate of state registration of the database No. 2018620743 in the Registry of databases on May 23, 2018). The data is available at the Local Ethics Committee (contact through the Chairperson of the Local Ethics Committee of the Far Eastern Scientific Center of Physiology and Pathology of Respiration, 22, Kalinina Street, 675000 Blagoveshchensk, Russian Federation) for researchers who meet the criteria for accessing confidential data.

**Funding:** The authors received no specific funding for this work.

**Competing interests:** The authors have declared that no competing interests exist.

of anxiety (56 persons, or 25.2%) turned out to be lower compared to rural residents (45 persons, or 34.8%; $\chi^2 = 34.08$; p<0.001).

## Conclusion

The burden of asthma introduces a greater imbalance in the health-related QoL of rural residents compared to urban residents in the Amur region of the Russian Federation. The absence of interrelationships of some QoL domains in rural residents suggested a disunity of the physical, psychological and social aspects of life. The rural residents suppress physical discomfort by the more frequent use of short bronchodilators. They often showed emotional instability with a predominance of anxiety, which affected the decrease in QoL in the psycho-emotional sphere.

## Introduction

Asthma is a common, chronic respiratory disease affecting 1–18% of the population in different countries [1]. Asthma imposes a significant burden on patients, healthcare system, and economies as a whole. Extensive research has shown the profound impact of adverse environmental, social, and behavioral conditions on human health, highlighting the importance of social determinants of health [2–4]. Epidemiological studies tend to show a greater prevalence of asthma in urban compared to rural populations [5]. In many parts of the world, rural–urban health disparities persist in terms of socio-demographic characteristics, access to healthcare, health status and prevalence of chronic diseases [5,6–10].

The subjective perception of health-related QoL has been considered of great relevance to measuring the course and outcomes of asthma [11]. QoL is also a valuable source of information on the causes of uncontrolled asthma and the prognosis of the disease [12].

In recent years, growing attention has been paid to a place of residence as a factor shaping health-related QoL. However, most of the studies on QoL originated in urban environments and few studies focused on comparing the perception of QoL in urban and rural areas [6,7,9,10,13]. Urban communities in the Russian Federation are more susceptible to social and psychological stress than rural communities, but they have more opportunities to undergo regular medical examinations [14]. Geographical remoteness of settlements, poor quality of road transportation links, and low availability of medical care are the common problems of rural communities in the Russian Federation [15].

The reason for the differences in asthma prevalence and the exacerbation rate in rural and urban areas may be due to the fact that populations have different lifestyles and cultures, as well as different environmental exposures [6,16,17]. The study of QoL can reveal disease-related and socially determined reasons for the existing possible differences in the course of asthma in urban and rural residents.

The purpose of this study was to assess the influence of permanent residence in urban and rural areas on health-related QoL in patients with mild-to-moderate asthma.

## Materials and methods

### Study cohort

The study sample included 351 patients aged 18 to 68 years (38.3% men and 61.7% women) with mild-to-moderate asthma. The sample consisted of 222 urban residents (1st group) and

**Table 1. The asthma duration.**

| Duration of asthma | Urban n = 222 | Rural n = 129 | $\chi^2$ ($p$) |
|---|---|---|---|
| < 5 years | 58 (26.1%) | 42 (32.6%) | 1.3561 (>0.05) |
| > 5 years | 164 (73.9%) | 87 (67.4%) | |

129 rural residents (2nd group). The duration of the disease in the sample ranged from 1.5 to 28 years. The distribution of patients in groups of urban and rural residents did not significantly differ in the duration of asthma (Table 1).

In the urban population, 17.1% attained a secondary education degree, 38.7% attained a vocational secondary education degree, 44.2% had a higher education degree, whereas in the rural population, these figures were 20.1%, 55.0% and 24.0%, respectively.

Clinical diagnosis of asthma was carried out in accordance with international recommendations [1]. The exclusion criteria included disabling comorbid pathologies and mental illnesses.

All patients, after preliminary familiarization with the study protocol, signed written informed consent. The study was carried out taking into account the requirements of the World Medical Association Declaration of Helsinki in compliance with the "Ethical principles for conducting scientific medical research with the involvement of humans" [18]. Approval was granted by the Local Biomedical Ethics Committee of the Far Eastern Scientific Center of Physiology and Pathology of Respiration (No. 132; 1 February 2019).

## Demographic and clinical variables

Subject-specific variables were collected by a self-designed questionnaire and included sex, age, body mass index, disease course, smoking index. Medication regimens were obtained by patient interviews, healthcare databases or prospectively from diary cards, and the frequency of each medication was recorded.

The smoking index (SI) was calculated as a number of cigarettes smoked per day × number of years of smoking / 20.

The degree of dyspnea was assessed using the modified Medical Research Council (mMRC) scale, the score of which ranges from 0 to 4 points, with higher scores indicating more severe dyspnea [19].

To assess disease control, patients completed the Asthma Control Test (ACT) [20]. The ACT survey is a patient-completed questionnaire with five items assessing asthma symptoms (daytime and nocturnal), use of rescue medications, and the effect of asthma on daily functioning. Each item includes five response options corresponding to a 5-point Likert-type rating scale. In scoring the ACT survey, responses for each of the five items are summed up to yield a score ranging from 5 (poor control of asthma) to 25 (complete control of asthma).

## Lung function outcome measurement

Participants completed spirometry using an Easy on-PC (nddMedizintechnik AG, Switzerland) according to the standard technique [21]. Lung function was measured as forced expiratory volume in the first second ($FEV_1$), forced vital capacity (FVC), peak expiratory flow (PEF), forced expiratory flow at 50% of FVC ($FEF_{50}$), forced expiratory flow at 75% of FVC ($FEF_{75}$). Percent predicted variables were calculated by dividing individual values by the ECCS predicted values for any person of a similar age, sex, race, and height. Ratio values of $FEV_1$/FVC were similarly translated as percent predicted.

## QoL assessment

All patients completed the self-reported questionnaires to measure QoL. To measure general QoL, the Medical Outcome Study Short-form 36-Item Health Survey (SF-36) was used, translated and validated into Russian [22,23]. This questionnaire included 36 questions that captured patients' perception of their health across eight domains of daily life: physical functioning (PF), role limitations due to physical health problems (RP), bodily pain (BP), general health (GH), vitality (VT), social functioning (SF), role limitations due to emotional problems (RE), mental health (MH) [22]. Responses to each question were transformed into SF-36 equivalent scores, and scores ranged from 0 to 100, with higher numerical scores indicating better QoL.

To measure specific health-related QoL, a standardized version of the Asthma Quality of Life Questionnaire (AQLQ) was used, translated and validated into Russian [24,25]. All cultural adaptations and linguistic validations of the questionnaire have been done by the MAPI Research Institute, Lyon, France (http://www.mapi-research.com). The AQLQ is a 32-item questionnaire used to assess the physical, occupational, emotional, and social qualities of adults aged 17 to 70 years with asthma. Patients respond to each question on a seven-point Likert scale with scores ranging from 1 (worst; severely impaired) to 7 (best; not impaired at all) and recall their experiences during the previous 2 weeks. Results are expressed as four domain scores (symptoms, 12 questions; activity limitation, 11 questions; emotional function, 5 questions; and environmental exposure, 4 questions) and as an overall score (general QoL, 32 questions).

## Anxiety and depression assessment

To determine anxiety and depression levels, the Hospital Anxiety and Depression Scale (HADS) was used [26]. This scale is divided into an anxiety subscale and a depression subscale, both containing seven intermingled items. Each item has 4 response categories ranging from 0 to 3. The both scales range from 0 to 21, with a higher score indicating a higher severity of anxiety and depression. For both anxiety and depression subscales, scores of 0–7 are considered normal, 8–10 are considered borderline abnormal (subclinical), and 11–21 are considered abnormal (clinical).

## Statistical analyses

Statistical analysis of the research results was carried out using standard methods of variation statistics, characterizing variation series for the normal distribution according to the Kolmogorov-Smirnov. In connection with the correspondence of the series to the normal distribution, the unpaired Student's t-test was used to determine the significance of differences between the mean values of the compared parameters. The significance level of $p < 0.05$ was taken into account. To find $p$ according to the unpaired criterion t, the degrees of freedom $f = n_1 + n_2 - 2$ was taken. The analysis of prevalence of a trait in the compared groups (frequency of alternative distribution) was carried out according to the $\chi^2$ test (Pearson's test). In order to determine the degree of connection between two random variables, Pearson correlation analysis was carried out, the correlation coefficient (r) and its significance were calculated.

## Results

### Characteristics of the study population

The clinical and functional characteristics of patients in the examined groups are presented in Table 2. The age and weight of patients in the compared groups did not differ significantly.

**Table 2. The clinical and functional characteristics of patients in the examined groups (mean±SE).**

| Variable | Urban n = 222 | $K_\alpha$* | Rural n = 129 | $K_\alpha$* | t value** | p value** |
|---|---|---|---|---|---|---|
| Age (years) | 33.7±0.7 | 0.513 | 34.9±0.8 | 0.730 | 1.1289 | 0.2937 |
| Weight (kg) | 74.9±1.2 | 0.571 | 73.3±1.4 | 0.588 | 0.8420 | 0.4077 |
| $FEV_1$ (% predicted) | 90.2±1.2 | 0.627 | 91.1±1.7 | 0.575 | 0.4325 | 0.6734 |
| $FEV_1$/FVC (% predicted) | 87.2±0.8 | 0.376 | 87.1±1.2 | 0.325 | 0.0496 | 0.9302 |
| PEF (% predicted) | 92.0±1.5 | 0.249 | 89.9±1.9 | 0.179 | 0.8675 | 0.3985 |
| $FEF_{50}$ (% predicted) | 64.0±1.6 | 0.116 | 64.1±2.5 | 0.637 | 0.0337 | 0.9837 |
| $FEF_{75}$ (% predicted) | 57.3±1.8 | 0.203 | 55.8±2.6 | 0.340 | 0.4743 | 0.6258 |
| The need for $\beta_2$-agonists (inh/day) | 2.7±0.3 | 0.883 | 4.2±0.6 | 0.890 | 2.2361 | 0.0221 |
| SI (pack/years) | 4.03±0.45 | 0.361 | 5.77±0.89 | 0.472 | 1.7606 | 0.0528 |
| mMRC (points) | 0.89±0.05 | 0.807 | 0.94±0.06 | 0.738 | 0.6402 | 0.4774 |
| (points) | 15.3±0.36 | 0.875 | 14.8±0.41 | 0.802 | 0.8156 | 0.4168 |

Notes.

* $K_\alpha$ - Kolmogorov–Smirnov test (critical value 0.895 at $\alpha = 0.05$)

** t-test; $FEV_1$: Forced expiratory volume in 1 second; FVC: Forced vital capacity; PEF: Peak expiratory flow; $FEF_{50}$: Forced expiratory flow at 50% of FVC; $FEF_{75}$:

Forced expiratory flow at 75% of FVC; SI: Smoking index; mMRC: Modified Medical Research Council; ACT: Asthma Control Test.

None of the measures of lung function showed significant differences between the groups of urban and rural residents, which corresponded to the absence of significant differences in the degree of dyspnea according to mMRC.

The study of the duration and intensity of smoking demonstrated that 57% of asthma patients in the urban areas and 46% in the rural areas smoked ($\chi^2 = 3.54$; p>0.05). The smoking index was not high—in urban residents, on average, it was 4.0±0.4 pack/years; in rural residents, it was 5.7±0.9 pack/years (p>0.05). The average age of respondents in both groups did not exceed 35 years old (33.7±0.7 vs. 34.9±0.8 years, respectively).

Although ACT did not differ significantly between groups, we noted a significantly higher need for $\beta_2$-agonists in rural areas compared to urban areas.

## Quality of life assessed by SF-36 and AQLQ scores

A larger number of points on the SF-36 scales among urban residents corresponded to a higher level of QoL (Table 3). However, comparative analysis revealed a significant difference only on the scale of emotional problems (RE), which was lower in patients from rural areas compared to urban residents. In the questionnaires, they characterized their own health as mediocre, with no either annual dynamics or worsening, noting frequent mood swings and moderate bodily pains. Both groups demonstrated low social activity.

Comparative analysis of QoL according to the AQLQ did not reveal significant differences (Table 4).

## Assessment of anxiety and depression

We did not observe any differences in the level of anxiety and depression on the HADS between groups of urban and rural residents (Table 5).

In the surveyed population, the average indicator of anxiety was at the subclinical level. The frequency of urban residents with a clinically significant level of anxiety (56 persons, or 25.2%) turned out to be lower compared to rural residents (45 persons, or 34.8%; $\chi^2 = 34.08$;

**Table 3. Indicators of the QoL in patients with asthma according to the SF-36 questionnaire (mean±SE).**

| Variable | Urban n = 222 | $K_\alpha$* | Rural n = 129 | $K_\alpha$* | t value** | p value** |
|---|---|---|---|---|---|---|
| PF | 73.2±1.43 | 0.846 | 71.7±1.76 | 0.653 | 0.6615 | 0.5040 |
| RP | 65.3±2.46 | 0.319 | 57.5±3.11 | 0.406 | 1.9670 | 0.0536 |
| BP | 73.3±1.79 | 0.844 | 71.0±2.50 | 0.733 | 0.7480 | 0.4558 |
| GH | 57.1±1.52 | 0.313 | 54.3±2.04 | 0.739 | 1.2006 | 0.2675 |
| VT | 60.9±1.40 | 0.534 | 66.2±5.75 | 0.219 | 1.1856 | 0.2703 |
| SF | 56.3±1.55 | 0.653 | 56.6±1.88 | 0.660 | 0.1231 | 0.7964 |
| RE | 70.4±2.30 | 0.406 | 59.4±3.10 | 0.728 | 3.8600 | 0.0042 |
| MH | 67.8±1.23 | 0.249 | 67.3±1.57 | 0.546 | 0.2507 | 0.7687 |

Notes.

* $K_\alpha$ - Kolmogorov–Smirnov test (critical value 0.895 at $\alpha$ = 0.05)

** t-test; PF—physical functioning; RP—role limitations due to physical health problems; BP—bodily pain; GH—general health; VT–vitality; SF—social functioning; RE—role limitations due to emotional problems; MH—mental health.

p<0.001). When analyzing individual questionnaires, we noted that the psycho-emotional background of urban respondents was stable, they were more confident in assessing their emotional state, defined it as positive and not interfering with social relations. Rural respondents exhibited emotional instability with a predominance of anxiety more frequently. Thus, 11% of urban and 20% of rural respondents answered "Most of the time" to the statement "I feel tense or 'wound up'". 42% of urban and 19% of rural residents answered "Definitely" to the statement "I can sit at ease and feel relaxed". The overall indicator on the depression scale did not exceed its normal level in either urban or rural residents.

## Correlation analysis data

In the group of urban residents, correlation analysis revealed the presence of positive correlations of SF-36 domains reflecting physical functioning (PF, RP, BP) with most scales of both QoL questionnaires (Table 6). The domains of social functioning (SF, VT) also had cross-dependencies both among themselves and with other QoL domains except the correlation between SF and RE. The general health (GH) positively correlated with most of the SF-36 scales except PF, and with the Activity limitation domain of the AQLQ. The domains of the emotional sphere (RE and MH) positively correlated with all scales of both QoL questionnaires among urban residents.

**Table 4. Indicators of the QoL in patients with asthma according to the AQLQ questionnaire (mean±SE).**

| Variable | Urban n = 222 | $K_\alpha$* | Rural n = 129 | $K_\alpha$* | t value** | p value** |
|---|---|---|---|---|---|---|
| Symptoms | 4.4±0.10 | 0.611 | 4.2±0.12 | 0.693 | 1.1604 | 0.3443 |
| Activity limitation | 4.3±0.09 | 0.605 | 4.2±0.10 | 0.759 | 1.2134 | 0.2645 |
| Emotion | 4.6±0.11 | 0.867 | 4.3±0.13 | 0.613 | 1.7052 | 0.1587 |
| Environment | 4.0±0.12 | 0.471 | 4.0±0.15 | 0.856 | 0.0520 | 0.8643 |
| General QoL | 4.4±0.09 | 0.772 | 4.3±0.09 | 0.815 | 0.7857 | 0.4227 |

Note.

* $K_\alpha$ - Kolmogorov–Smirnov test (critical value 0.895 at $\alpha$ = 0.05)

** t-test.

**Table 5. Indicators of anxiety and depression in patients with asthma according to HADS (mean±SE).**

| Variable | Urban n = 222 | $K_\alpha$* | Rural n = 129 | $K_\alpha$* | t value** | ** |
|---|---|---|---|---|---|---|
| Anxiety scale | 7.2±0.29 | 0.876 | 7.7±0.35 | 0.690 | 1.1563 | 0.2758 |
| Depression scale | 4.9±0.25 | 0.263 | 5.4±0.32 | 0.752 | 1.1824 | 0.2707 |

Note.

* $K_\alpha$ - Kolmogorov–Smirnov test (critical value 0.895 at $\alpha = 0.05$)

atest.

In the group of rural residents, a comparative analysis showed the absence of the correlation of PF with VT and SF (Table 7). Noteworthy is the absence of significant correlations of BP, GH, SF and VT from most QoL scales. Social functioning (SF) positively correlated only with the scale of general health (GH). In this group, we noted a clear effect of emotional expression (RE) on the self-assessment of the physical status of QoL according to the PF and RP, but not on the domains of psychosocial well-being. In the group of rural patients, in contrast to urban residents, correlation analysis showed a total lack of relationship between the Activity limitation and all QoL scales (Table 7).

In the group of urban residents, in contrast to rural residents, correlation analysis showed a negative relationship of smoking (SI) with GH, PF and RP. Dyspnea (according to the mMRC) was involved in the formation of specific QOL assessment in all AQLQ domains, as well as in 5

**Table 6. Correlation matrix of the quality of life variables in the group of urban residents.**

| Variable | RP | BP | GH | VT | SF | RE | MH | A | S | EF | E | GQoL |
|---|---|---|---|---|---|---|---|---|---|---|---|---|
| PF | r = 0.45 p<0.0001 | r = 0.42 p<0.0001 | r = 0.01 p = 0.8642 | r = 0.24 p = 0.0007 | r = 0.18 p = 0.0111 | r = 0.53 p<0,0001 | r = 0.22 p = 0.0018 | r = 0.60 p<0.0001 | r = 0.52 p<0.0001 | r = 0.46 p<0.0001 | r = 0.54 p<0.0001 | r = 0.59 p<0.0001 |
| RP | | r = 0.45 p<0.0001 | r = 0.22 p = 0.0018 | r = 0.30 p<0.0001 | r = 0,12 p = 0.0982 | r = 0.39 p<0.0001 | r = 0.16 p = 0.0226 | r = 0.52 p<0.0001 | r = 0.35 p<0.0001 | r = 0.29 p<0.0001 | r = 0.30 p<0.0001 | r = 0.38 p<0.0001 |
| BP | | | r = 0.26 p = 0.0002 | r = 0.33 p<0.0001 | r = 0.09 p = 0.2332 | r = 0.45 p<0.0001 | r = 0.28 p = 0.0001 | r = 0.36 p<0.0001 | r = 0.31 p<0.0001 | r = 0.26 p = 0.0002 | r = 0.19 p = 0.0085 | r = 0.26 p = 0.0002 |
| GH | | | | r = 0.19 p = 0.0085 | r = 0.16 p = 0.0226 | r = 0.16 p = 0.0226 | r = 0.18 p = 0.0111 | r = 0.16 p = 0.0294 | r = -0.03 p = 0.6758 | r = 0.05 p = 0.5011 | r = 0.13 p = 0.0625 | r = 0.02 p = 0.8209 |
| VT | | | | | r = 0.26 p = 0.0002 | r = 0.29 p<0.0001 | r = 0.22 p = 0.0018 | r = 0.18 p = 0.0111 | r = 0.17 p = 0.0179 | r = 0.22 p = 0.0018 | r = 0.21 p = 0.0028 | r = 0.23 p = 0.0014 |
| SF | | | | | | r = 0.10 p = 0.1733 | r = 0.22 p = 0.0018 | r = 0.18 p = 0.0111 | r = 0.17 p = 0.0179 | r = 0.22 p = 0.0018 | r = 0.21 0.0028 | r = 0.23 p = 0.0014 |
| RE | | | | | | | r = 0.41 p<0.0001 | r = 0.49 p<0.0001 | r = 0.28 p = 0.0001 | r = 0.35 p<0.0001 | r = 0.42 p<0.0001 | r = 0.34 p<0.0001 |
| MH | | | | | | | | r = 0.34 p<0.0001 | r = 0.25 p = 0.0005 | r = 0.20 p = 0.0060 | r = 0.29 p<0.0001 | r = 0.27 p = 0.0002 |
| A | | | | | | | | | r = 0.67 p<0.0001 | r = 0.65 p<0.0001 | r = 0.67 p<0.0001 | r = 0.81 p<0.0001 |
| S | | | | | | | | | | r = 0.66 p<0.0001 | r = 0.56 p<0.0001 | r = 0.80 p<0.0001 |
| EF | | | | | | | | | | | r = 0.59 p<0.0001 | r = 0.75 p<0.0001 |
| E | | | | | | | | | | | | r = 0.71 p<0.0001 |

Notes. PF—physical functioning; RP—role limitations due to physical health problems; BP—bodily pain; GH—general health; VT–vitality; SF—social functioning; RE—role limitations due to emotional problems; MH—mental health; A—activity limitation; S–symptoms; EF—emotional function; E—environmental exposure; GQoL—general QoL.

**Table 7. Correlation matrix of the quality of life variables in the group of rural residents.**

| Variable | RP | BP | GH | VT | SF | RE | MH | A | S | EF | E | GQoL |
|---|---|---|---|---|---|---|---|---|---|---|---|---|
| **PF** | r = 0.30 p = 0.0008 | r = 0.32 p = 0.0003 | r = 0.06 p = 0.5355 | r = 0.17 p = 0.0725 | r = 0.02 p = 0.8694 | r = 0.35 p<0.0001 | r = 0.21 p = 0.0227 | r = -0.03 p = 0.7417 | r = 0.31 p = 0.0007 | r = 0.36 p = 0.0001 | r = 0.34 p = 0.0002 | r = 0.37 p<0.0001 |
| **RP** | | r = 0.26 p = 0.0049 | r = 0.34 p = 0.0002 | r = 0.18 p = 0.0484 | r = 0.10 p = 0.2981 | r = 0.45 p<0.0001 | r = -0.02 p = 0.8694 | r = 0.01 p = 0.9698 | r = 0.20 p = 0.0349 | r = 0.32 p = 0.0006 | r = 0.31 p = 0.0006 | r = 0.33 p = 0.0003 |
| **BP** | | | r = 0.12 p = 0.1917 | r = 0.14 p = 0.1436 | r = 0.10 p = 0.2981 | r = 0.06 p = 0.5458 | r = 0.10 p = 0.2981 | r = -0.05 p = 0.6441 | r = 0.20 p = 0.0277 | r = 0.09 p = 0.3354 | r = 0.36 p = 0.0001 | r = -0.08 p = 0.4123 |
| **GH** | | | | r = 0.14 p = 0.1436 | r = 0.19 p = 0.0415 | r = 0.08 p = 0.3777 | r = 0.12 p = 0.2086 | r = -0.11 p = 0.2268 | r = 0.03 p = 0.7579 | r = 0.02 p = 0.8694 | r = -0.02 p = 0.8694 | r = 0.15 p = 0.0990 |
| **VT** | | | | | r = 0.02 p = 0.8694 | r = 0.17 p = 0.0690 | r = 0.15 p = 0.0990 | r = 0.17 p = 0.0690 | r = 0.11 p = 0.2268 | r = 0.12 p = 0.2086 | r = 0.15 p = 0.0990 | r = 0.15 p = 0.0990 |
| **SF** | | | | | | r = 0.05 p = 0.6441 | r = 0.05 p = 0.6441 | r = 0.09 p = 0.2272 | r = -0.02 p = 0.8694 | r = 0.13 p = 0.1671 | r = 0.13 p = 0.1671 | r = 0.02 p = 0.8694 |
| **RE** | | | | | | | r = 0.10 p = 0.2981 | r = 0.15 p = 0.0990 | r = 0.26 p = 0.0043 | r = 0.17 p = 0.0690 | r = 0.27 p = 0.0036 | r = 0.29 p = 0.0014 |
| **MH** | | | | | | | | r = 0.09 p = 0.2272 | r = 0.28 p = 0.0028 | r = 0.31 p = 0.0008 | r = 0.25 p = 0.0062 | r = 0.30 p = 0.0011 |
| **A** | | | | | | | | | r = 0.05 p = 0.6441 | r = 0.15 p = 0.0990 | r = -0.11 p = 0.2268 | r = -0.03 p = 0.7417 |
| **S** | | | | | | | | | | r = 0.63 p<0.0001 | r = 0.57 p<0.0001 | r = 0.78 p<0.0001 |
| **EF** | | | | | | | | | | | r = 0.49 p<0.0001 | r = 0.76 p<0.0001 |
| **E** | | | | | | | | | | | | r = 0.72 p<0.0001 |

Notes. PF—physical functioning; RP—role limitations due to physical health problems; BP—bodily pain; GH—general health; VT–vitality; SF—social functioning; RE—role limitations due to emotional problems; MH—mental health; A—activity limitation; S–symptoms; EF—emotional function; E—environmental exposure; GQoL—general QoL.

out of 8 domains of the SF-36 (PF, RP, BP, RE, MH) in urban residents (Table 8). In rural residents, we found similar positive correlations of the mMRC with the "Symptoms", "Emotion", "Environment", and "General QoL" domains, as well as with three SF-36 domains (PF, VT, MH) (Table 8).

Asthma control was positively correlated with PF and MH in both the urban and rural cohorts. In urban residents, asthma control also clearly negatively depended on smoking (r = -0.19; p = 0.0089), anxiety (r = -0.24; p = 0.0101) and depression (r = -0.30; p = 0.0001).

Correlation analysis revealed significant negative dependences of all SF-36 and AQLQ scales on the level of anxiety and depression in the cohort of urban asthma patients, except for the GH (Table 8). In the cohort of rural residents, the correlations between anxiety and depression with the domains RP, VT, SF, and "Activity limitation" disappeared (Table 8).

Fig 1 illustrates the difference in the number of statistically significant positive and negative correlations of QoL indicators according to the SF-36 and AQLQ questionnaires with the asthma control level (ACT) and the smoking index (SI). Among the asthma patients living in urban areas, the total number of significant correlations of the studied set of QoL variables was more than twice as high as in rural residents (84 correlations vs. 38 correlations, respectively).

## Discussion

Previously, a lower level of QoL was demonstrated in elderly people living in rural areas [9], patients with hypertension [8], oral health conditions [7]. The results of this study indicate

**Table 8. Correlations of the SF-36 and AQLQ scales with smoking index, mMRC, Asthma Control Test, anxiety and depression according to HADS (mean±SE).**

| Variable | SI | | mMRC | | ACT | | Anxiety | | Depression | |
|---|---|---|---|---|---|---|---|---|---|---|
| | urban | rural | urban | rural | urban | rural | urban | rural | urban | rural |
| PF | r = -0.23 p = 0,0014 | NS | r = -0.42 p<0.0001 | r = -0.21 p = 0,0220 | r = 0.46 p<0.0001 | r = 0.25 p = 0.0063 | r = -0.45 p<0.0001 | r = -0.40 p<0.0001 | r = -0.45 p<0.0001 | r = -0.48 p<0.0001 |
| RP | r = -0.21 p = 0,0041 | NS | r = -0.16 p = 0.0253 | NS | r = 0.32 p<0.0001 | NS | r = -0.20 p = 0.0055 | NS | r = -0.36 p<0.0001 | NS |
| BP | NS | NS | r = 0.17 p = 0.0149 | NS | r = 0.29 p<0.0001 | NS | r = -0.33 p<0.0001 | r = -0.20 p = 0.0289 | r = -0.43 p<0.0001 | NS |
| GH | r = -0.16 p = 0.0312 | NS | NS | NS | NS | NS | NS | NS | NS | NS |
| VT | NS | NS | NS | r = -0.20 p = 0,0287 | NS | NS | r = -0.31 p<0.0001 | NS | r = -0.35 p<0.0001 | NS |
| SF | NS | NS | NS | NS | r = 0.21 p = 0.0026 | NS | r = -0.16 p = 0.0212 | NS | r = -0.17 p = 0.0182 | NS |
| RE | NS | NS | r = -0.34 p<0.0001 | NS | r = 0.39 p<0.0001 | NS | r = -0.42 p<0.0001 | r = -0.21 p = 0.0238 | r = -0.52 p<0.0001 | r = -0.26 p = 0.0059 |
| MH | NS | NS | r = -0.17 p = 0.0149 | r = -0.19 p = 0,0383 | r = 0.23 p = 0.0014 | r = 0.20 p = 0.0347 | r = -0.50 p<0.0001 | r = -0.40 p<0.0001 | r = -0.37 p<0.0001 | r = -0.30 p = 0.0013 |
| A | NS | NS | r = -0.32 p<0.0001 | NS | r = 0.47 p<0.0001 | NS | r = -0.40 p<0.0001 | r = -0.38 p<0.0001 | r = -0.48 p<0.0001 | r = -0.46 p<0.0001 |
| S | NS | NS | r = -0.22 p = 0.0027 | r = -0.43 p<0,0001 | r = 0.49 p<0.0001 | r = 0.440 p<0.0001 | r = -0.31 p<0.0001 | r = -0.32 p = 0.0005 | r = -0.32 p<0.0001 | r = -0.33 p = 0.0004 |
| EF | NS | NS | r = -0.30 p = 0.0027 | r = -0.30 p = 0.0013 | r = 0.44 p<0.0001 | r = 0.35 p = 0.0001 | r = -0.35 p<0.0001 | r = -0.37 p = 0.0001 | r = -0.37 p<0.0001 | r = -0.39 p<0.0001 |
| E | NS | NS | r = -0.32 p<0.0001 | r = -0.32 p = 0.0040 | r = 0.46 p<0.0001 | r = 0.38 p<0.0001 | r = -0.26 p = 0.0002 | r = -0.33 p = 0.0003 | r = -0.33 p<0.0001 | r = -0.37 p<0.0001 |
| GQoL | NS | NS | r = -0.29 p = 0.0001 | r = -0.46 p<0.0001 | r = 0.54 p<0.0001 | r = 0.45 p<0.0001 | r = -0.29 p<0.0001 | r = -0.36 p = 0.0001 | r = -0.31 p<0.0001 | r = -0.42 p<0.0001 |

Notes. PF—physical functioning; RP—role limitations due to physical health problems; BP—bodily pain; GH—general health; VT–vitality; SF—social functioning; RE—role limitations due to emotional problems; MH—mental health; A—activity limitation; S–symptoms; EF—emotional function; E—environmental exposure; GQoL—general QoL; SI–smoking index; mMRC–modified Medical Research Council; ACT–Asthma Control Test.

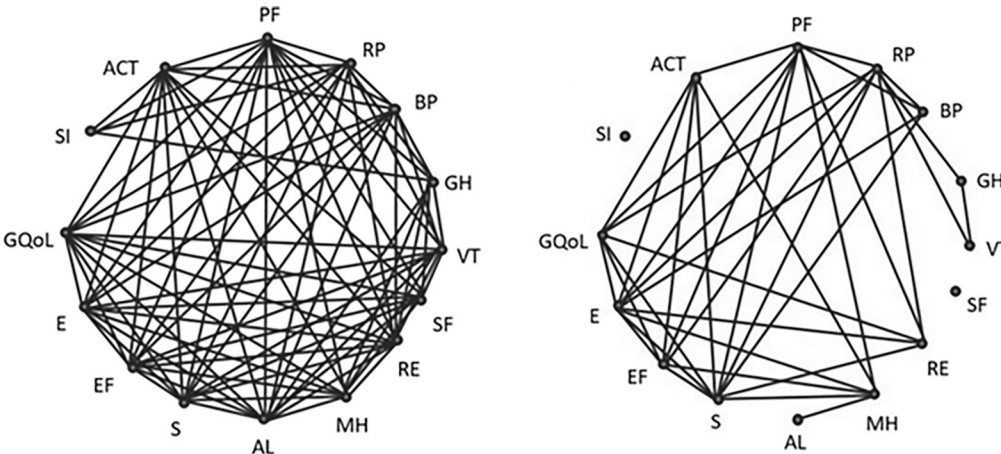

**Fig 1. A diagram of correlations of QoL domains according to the SF-36 and AQLQ questionnaires, smoking index and asthma control.** The group of urban residents is on the left hand side, a group of rural residents is on the right hand side. PF: Physical functioning; RP: Role physical; BP: Bodily pain; GH: General health; VT: Vitality; SF: Social functioning; RE: Role emotional; MH: Mental health; AL: Activity limitation; S: Symptoms; EF: Emotion functioning; E: Environment; GQoL: General quality of life; SI: Smoking index; ACT: Asthma control test.

that the place of permanent residence makes a contribution to the formation of the QoL in asthma patients living in urban and rural areas. The burden of disease had a more significant impact on the QoL of rural residents. One of the important reasons for these differences is considered the lower level of availability of healthcare resources in rural areas of the Russian Federation [20]. Rural patients with asthma are more often unable to comprehensively assess their current health and the risks for its deterioration. On the contrary, a relatively high level of medical awareness and readiness for health preservation results in a better QoL of urban residents with asthma.

As follows from Fig 1, the correlation analysis demonstrated a significant depletion of the field of correlations in rural residents with asthma compared to urban residents. The absence of interrelationships of some QoL domains in rural residents suggested a disunity of the physical, psychological and social aspects of life, which negatively affects the QoL level among asthma patients.

Anxiety and depression are common and relevant comorbidities in outpatients with asthma, and are associated with an uncontrolled course of the disease and decreased QoL [27–29]. There was a significant inverse correlation between the overall assessment of the quality of life and the average anxiety scores in asthma [29]. In the present study, psycho-emotional characteristics shaped the QoL of asthma patients in different ways, depending on the place of residence. Among urban residents, mild anxiety can be viewed as a normative phenomenon that accompanies patients' adaptation to a chronic illness. Low anxiety is known to provide predictive competence, thus performing an adaptive function. Anxiety is viewed not only as an emotional phenomenon, but also includes cognitive and motivational components that shape behavior. Correlations with QoL domains indicated that a low level of anxiety stimulates urban residents to maintain a decent level of QoL by means of searching for optimal opportunities to live in a society, despite the existing informational and intellectual overloads. At the same time, among rural residents, anxiety can act as one of the main factors of psychological disadaptation. Rural respondents demonstrated emotional instability with a predominance of anxiety and mental indifference more frequently, as evidenced by the absence of a number of correlations with QoL domains.

The smoking factor in urban residents had a significant impact on both the indices of some QoL domains and the level of ACT, in contrast to rural residents. Previous studies have shown a statistically significant association between smoking and QoL [30–32]. At the same time, such a relationship was not found among rural residents in Japan [33]. In our study, the absence of correlations between the smoking index and the QoL in the rural residents suggested that that they do not recognize smoking as a trigger factor for feeling unwell due to a lack of communication with a doctor, low awareness and availability of healthcare, all of which are typical for rural areas in the Amur region of the Russian Federation.

Based on the output of the correlation analysis, we noted that dyspnea plays a more significant role in the assessment of specific QoL in urban residents. Perhaps this is due to the reported lower frequency of use of short-acting $\beta_2$-agonists by urban residents.

Poor asthma control, along with older age and a lower level of education, has been noted as one of the factors contributing to the QoL decline [34]. In our study, asthma control in urban and rural residents was low and did not differ significantly in the studied groups. In both groups, it influenced both the level of physical functioning and mental health according to the SF-36 scales. In turn, the level of asthma control in urban residents, in contrast to rural residents, significantly depended on the factor of smoking and emotional disorders.

Some limitations should be taken into account in the interpretation of the results. Firstly, the findings may not be representative of the entire country, as the study only includes patients living in Amur Region of Russian Federation. Furthermore, study sample does not include the

patients with severe asthma that limit the accuracy of the inference for the asthma in general. Data on comorbidities in the study sample are limited. Despite these shortcomings, this study highlights some characteristics shaping the urban-rural disparities of health-related quality of life in patients with asthma living in the Russian Federation.

## Conclusions

The burden of asthma introduces a greater imbalance in the health-related QoL of rural residents compared to urban residents in the Amur region of the Russian Federation. The absence of interrelationships of some QoL domains in rural residents suggested a disunity of the physical, psychological and social aspects of life. The rural residents suppress physical discomfort by the more frequent use of short bronchodilators. They often showed emotional instability with a predominance of anxiety, which affected the decrease in QoL in the psycho-emotional sphere.

## Author Contributions

**Conceptualization:** Natalia L. Perelman, Victor P. Kolosov.

**Data curation:** Natalia L. Perelman.

**Formal analysis:** Natalia L. Perelman.

**Investigation:** Natalia L. Perelman.

**Methodology:** Natalia L. Perelman, Victor P. Kolosov.

**Writing – original draft:** Natalia L. Perelman.

**Writing – review & editing:** Victor P. Kolosov.

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
