## [Decision Letter · Decision Letter 0]

18 Jul 2022

PONE-D-22-00706Differences in the health-related quality of life in patients with asthma living in urban and rural areas in the Russian FederationPLOS ONE

Dear Dr. Perelman,

Thank you for submitting your manuscript to PLOS ONE. After careful consideration, we feel that it has merit but does not fully meet PLOS ONE’s publication criteria as it currently stands. Therefore, we invite you to submit a revised version of the manuscript that addresses the points raised during the review process.

We look forward to receiving your revised manuscript.

Kind regards,

Munn-Sann Lye, MBBS, MPH, DrPH

Academic Editor

PLOS ONE

Journal Requirements:

**
https://journals.plos.org/plosone/s/file?id=ba62/PLOSOne_formatting_sample_title_authors_affiliations.pdf
**

Reviewers' comments:

Reviewer's Responses to Questions

**Comments to the Author**

1. Is the manuscript technically sound, and do the data support the conclusions?

Reviewer #1: Partly

Reviewer #2: Partly

2. Has the statistical analysis been performed appropriately and rigorously? 

Reviewer #1: Yes

Reviewer #2: Yes

3. Have the authors made all data underlying the findings in their manuscript fully available?

Reviewer #1: Yes

Reviewer #2: Yes

4. Is the manuscript presented in an intelligible fashion and written in standard English?

Reviewer #1: Yes

Reviewer #2: No

5. Review Comments to the Author

Reviewer #1: Introduction section should be shortened. All statements should be followed by corresponding references. All statements without references should be deleted. Exact region where patients live should be mentioned instead of the country name as it was a small regional study. Explanatory data on each of used questionnaire should be provided in methods section. Information of Russian translation and validation of used instruments should also be provided. All abbreviations should be explained when first mentioned. It is unclear how data of rural patients was collected. Was it collected by rural medical authorities? Lowe scores of SF-36 mean better QoL. Therefore it is important each time clearly indicate if authors mean higher or worse QoL or SF-36 scores. Conclusions should be based exclusively on study results.

Tables should be in the same style, all abbreviations should be explained.

Minor comment: “with no annual dynamics” – how dynamics was studied if patients filled in questionnaires only once?

Reviewer #2: In the study under review, the authors have examined the health-related quality of life in patients with asthma in Russian Federation as a function of their residence, i.e., rural versus urban. Although the current study is important, I have some reservations regarding issues that I noticed while reading through the article. Particularly, the major issue is related to the Materials and Methods and Results sections. Also, the entire text requires careful editing for English syntax and grammar. I have outlined those below point-wise.

Abstract:

Conclusion is not a conclusion, but more a summary. Please write an appropriate conclusion, which is supported by the important outcomes of the study.

Introduction:

Page 3, Line 50. The first statement needs rephrasing.

Page 4, Lines 73-75: “Information and intellectual overloads cause ……….. negatively affects the course and prognosis of a chronic disease.” Reference missing.

Page 4, Lines 75-78: “Geographical remoteness of settlements, poor quality …………. rural communities in the Russian Federation.” How this statement is relevant to the current study?

Page 4, Lines 78-83: “In addition, rural residents suffer from declining …………… give patients recommendations for a healthy lifestyle and behavior.” The authors did not provide any supporting references in support of these statements.

Page 4, Lines 94-96: “The purpose of this study is to assess the influence of social factors on ……………… permanently residing in urban and rural areas. What are the social factors that the Authors have considered for the current study? It should be explained here.

The Introduction section is very vague. It should be focused and precise and should be written keeping in mind the objectives of the study.

Materials & Methods:

This section is not written properly. It needs drastic revision.

Study cohort

Page 5, Lines 100-101: “The duration of the disease in the 101 sample ranged from 1.5 to 28 years.” The Authors should also have examined the impact of the ‘duration of the disease’ on the QoL. Does ‘the duration of the disease’ similar between urban and rural residents?

Outcome measures and explanatory variables

Spirometry outcome variables should be explained.

All the questionnaires should be explained in detail, such as the number of items, scoring, interpretation of scores, etc.

Statistical analyses:

Results of Kolmogorov-Smirnov and Pearson-Mises should have been given that reflect the normality of the data.

Results:

Table 1: In the column title ‘urban’ and ‘rural’ should have been written in place of ‘1 group’ and ‘2 group.’ Abbreviated variables in the 1st column should be spelled out in the footnote. t-value and p-value should be given in a separate column.

Correlation results should be given separately in the main text in a separate sub-heading and values should be given in a Table.

Quality of life assessed by AQLQ score

Page 11, Lines 181-182: “However, in the urban residents group, the level of “Activity limitation” correlated with all domains of the SF-36 questionnaire.” The authors did not specify whether the correlation is positive or negative.

Figure 1 is not clear.

Page 12, Lines 194-196: “Among the asthma patients living in urban areas, …………. significantly higher than in rural residents (84 vs. 38).” What is 84 vs. 38? The Correlation result is not written properly. Correlation coefficient values are not given.

Page 12, Lines 204-205: “We did not observe any differences …………… on the HADS.” The authors did not mention the “difference between which group.”

Page 12, Line 206: ‘subclinical level;’ what does it mean?

Page 12, Lines 206 and 207: What is the difference between subclinical level and clinical level?

Results of HADS should be given in a Table.

Pages 12 and 13, Lines 207-209: “The number of urban residents with a ……… rural residents (34.8%; χ²=34.08; p<0.001). What is the frequency of urban and rural residents with clinically significant levels of anxiety?

Page 13, Lines 209-213: “When analyzing individual 209 questionnaires, we noted that the psycho-emotional background ……………… instability with a predominance of anxiety more frequently.” There is no data/results in support of these statements.

Page 13, Lines 215-219: “In the urban residents group, the correlation analysis …………….. correlations of anxiety with the RP, VT, SF, “Activity limitation” domains disappeared.” Again, there is no data/results in support of these statements.

All results of the Student's t-test, χ² test, and Pearson correlation analysis should be given in Tables or Figures. Also, the result of Kolmogorov-Smirnov and Pearson-Mises criteria should be described.

Abbreviated variables in all the Tables should be spelled out in the footnote.

Overall, this section is not written properly. It needs drastic revision.

Discussion:

The main findings of the current study have not been discussed properly in this section.

Page 16, Lines 286-288: “In our study, the absence of correlations between the smoking index and the QoL in the rural residents suggested their riskier behavior in relation to their own health.” This reviewer is unable to understand the above statement.

Page 16, Lines 294-296: “One of the explanations may lie in the varying impact of environmental factors on rural and urban residents, domestic animal ownership, as well as lower intake of controller medications.” This statement is not clear to me.

Conclusion:

The conclusions however let it down as this is more of a summary than a conclusion. Please write an appropriate conclusion, which is supported by the important outcomes of the study.

6. PLOS authors have the option to publish the peer review history of their article (what does this mean?). If published, this will include your full peer review and any attached files.

Reviewer #1: No

Reviewer #2: No

---

## [Author Response · Author response to Decision Letter 0]

28 Oct 2022

Response to reviewers

Dear Reviewer #1,

We thank you for your great work in reviewing our manuscript and for your comments. Below are the responses to the comments and information about the corrections made:

• Introduction section should be shortened. All statements should be followed by corresponding references. All statements without references should be deleted. Exact region where patients live should be mentioned instead of the country name as it was a small regional study.

Response: In accordance with the comments, we have shortened the "Introduction" section. All statements are followed by corresponding references. All unlinked statements have been deleted. The country name in the title and text of the article is supplemented with a mention of the specific region in which the study was conducted (Amur Region of the Russian Federation).

• Explanatory data on each of used questionnaire should be provided in methods section. Information of Russian translation and validation of used instruments should also be provided.

Response: Now the “Materials and Methods” section provides complete information on each of the questionnaires, as well as information on the translation into Russian and validation of the tools used with the appropriate links.

• All abbreviations should be explained when first mentioned.

Response: In accordance with the remark, all abbreviations are explained when first mentioned and in the notes under tables and figure. 

• It is unclear how data of rural patients was collected. Was it collected by rural medical authorities?

Response: We clarify that all patients were examined and interviewed during a consultation appointment at the consultative and diagnostic department of the Far Eastern Scientific Center of Physiology and Pathology of Respiration. Each patient completed the questionnaire independently in the presence of a health worker. Rural medical authorities only sent patients for consultation.

• Lowe scores of SF-36 mean better QoL. Therefore, it is important each time clearly indicate if authors mean higher or worse QoL or SF-36 scores.

Response: You are correct in pointing out that higher Lowe scores mean higher QoL. We reflected this in the first sentence of the “Quality of life assessed by SF-36 and AQLQ scores” section (Page 11): “A larger number of points on the SF-36 scales among urban residents corresponded to a higher level of QoL…”

• Conclusions should be based exclusively on study results.

Response: We corrected the conclusions in accordance with the results of the study.

• Tables should be in the same style, all abbreviations should be explained.

Response: The tables are unified. All abbreviations are explained in the notes under the tables.

• Minor comment: “with no annual dynamics” – how dynamics was studied if patients filled in questionnaires only once?

Response: Patients filled out questionnaires once, indeed. The mention in the text about the annual dynamics is not the result of a repeated study, but only an analysis of the answers to the second question of the SF-36 questionnaire: “Compared to one year ago, how would you rate your health in general now?”

Once again, thank you for your positive review and comments.

Sincerely,

Dr. Natalia M. Perelman

 

Dear Reviewer #2,

We thank you for your great work in reviewing our manuscript and for your comments. Below are the responses to the comments and information about the corrections made:

• Abstract: Conclusion is not a conclusion, but more a summary. Please write an appropriate conclusion, which is supported by the important outcomes of the study.

Response: We corrected the conclusions in accordance with the results of the study.

• Introduction: Page 3, Line 50. The first statement needs rephrasing.

Response: We have rephrased the statement (Page 3, Lines 55-57).

• Page 4, Lines 73-75: “Information and intellectual overloads cause ……….. negatively affects the course and prognosis of a chronic disease”. Reference missing.

Response: We have removed this sentence from the text.

• Page 4, Lines 75-78: “Geographical remoteness of settlements, poor quality …………. rural communities in the Russian Federation.” How this statement is relevant to the current study?

Response: We have removed this sentence from the text.

• Page 4, Lines 78-83: “In addition, rural residents suffer from declining …………… give patients recommendations for a healthy lifestyle and behavior.” The authors did not provide any supporting references in support of these statements.

Response: We have removed this sentence from the text.

• Page 4, Lines 94-96: “The purpose of this study is to assess the influence of social factors on ……………… permanently residing in urban and rural areas. What are the social factors that the Authors have considered for the current study? It should be explained here.

Response: We have made a more correct description of the purpose of this study (Page 4, Lines 82-84).

• The Introduction section is very vague. It should be focused and precise and should be written keeping in mind the objectives of the study.

Response: We have shortened and corrected the "Introduction" section.

• Page 5, Lines 100-101: “The duration of the disease in the 101 sample ranged from 1.5 to 28 years.” The Authors should also have examined the impact of the ‘duration of the disease’ on the QoL. Does ‘the duration of the disease’ similar between urban and rural residents?

Response: We did not set ourselves the task of studying the effect of “disease duration” on QoL. However, to exclude such speculations, we additionally provided Table 1, in which we showed the same distribution of patients with different duration of illness in groups of urban and rural residents.

• Spirometry outcome variables should be explained.

Response: We have added a detailed description of the spirometry outcome variables (Page 7, Lines 130-137).

• All the questionnaires should be explained in detail, such as the number of items, scoring, interpretation of scores, etc.

Response: We have added a detailed description of all questionnaires.

• Statistical analyses: Results of Kolmogorov-Smirnov and Pearson-Mises should have been given that reflect the normality of the data.

Response: We entered into the tables the values of the Kolmogorov-Smirnov criterion, from which the data normal distribution follows. The Pearson-Mises criterion has been removed from the text, since this criterion duplicates the results of assessing the distribution normality using the Kolmogorov-Smirnov criterion.

• Table 1: In the column title ‘urban’ and ‘rural’ should have been written in place of ‘1 group’ and ‘2 group.’ Abbreviated variables in the 1st column should be spelled out in the footnote. t-value and p-value should be given in a separate column.

Response: We have made all the corrections indicated (see Table 2 on Page 10)

• Correlation results should be given separately in the main text in a separate sub-heading and values should be given in a Table.

Response: We have separated the correlation results into the subsection “Correlation analysis data” (Page 14), providing it with Tables 6-8, which contain comprehensive information about the correlation coefficients and their significance.

• Page 11, Lines 181-182: “However, in the urban residents group, the level of “Activity limitation” correlated with all domains of the SF-36 questionnaire.” The authors did not specify whether the correlation is positive or negative.

Response: We have additionally indicated whether there are positive or negative correlations in all cases.

• Figure 1 is not clear.

Response: The figure reflects statistically significant correlations between the domains of the SF-36 and AQLQ questionnaires, as well as the smoking index and ACT. It clearly demonstrates a significant depletion of the field of correlations in rural residents with asthma compared to urban residents. In our opinion, the disappearance of interrelationships of some QoL domains in rural residents suggested a disunity of the physical, psychological and social aspects of life, which negatively affects the QoL level among asthma patients. These explanations are given on Pages 15-16 (Lines 284-289) and Page 22 (Lines 327-332).

• Page 12, Lines 194-196: “Among the asthma patients living in urban areas, …………. significantly higher than in rural residents (84 vs. 38).” What is 84 vs. 38? The Correlation result is not written properly. Correlation coefficient values are not given.

Response: We have changed this sentence (Page 16, Lines 287-289). It reflects in text form the above information about a significantly smaller number of statistically significant correlations of QoL domains with each other in rural residents compared to urban residents. 84 significant correlations were found among urban residents, and only 38 among rural residents.

• Page 12, Lines 204-205: “We did not observe any differences …………… on the HADS.” The authors did not mention the “difference between which group.”

Response: We explained which groups did not show differences: “We did not observe any differences in the level of anxiety and depression on the HADS between groups of urban and rural residents” (Page 13, lines 229-230)

• Page 12, Line 206: ‘subclinical level;’ what does it mean?

Page 12, Lines 206 and 207: What is the difference between subclinical level and clinical level?

Response: We have given the necessary explanations in the section “Methods” (subsection “Anxiety and depression assessment”, Page 8, Lines 167-169).

• Results of HADS should be given in a Table.

Response: HADS results are listed in a separate Table 5 (Page 13)

• Pages 12 and 13, Lines 207-209: “The number of urban residents with a ……… rural residents (34.8%; χ²=34.08; p<0.001). What is the frequency of urban and rural residents with clinically significant levels of anxiety?

Response: We corrected the sentence (Page 13, Lines 236-238): “The frequency of urban residents with a clinically significant level of anxiety (56 persons, or 25.2%) turned out to be lower compared to rural residents (45 person, or 34.8 %; χ²=34.08; p<0.001)

• Page 13, Lines 209-213: “When analyzing individual 209 questionnaires, we noted that the psycho-emotional background ……………… instability with a predominance of anxiety more frequently.” There is no data/results in support of these statements.

Response: To support this, we added the data on the frequency of individual responses to the HADS statements “I feel tense or 'wound up'” and “I can sit at ease and feel relaxed” (Page 14, Lines 243-246).

• Page 13, Lines 215-219: “In the urban residents group, the correlation analysis …………….. correlations of anxiety with the RP, VT, SF, “Activity limitation” domains disappeared.” Again, there is no data/results in support of these statements.

Response: We moved this statement into a separate subsection “Correlation analysis data” (Page 15, Lines 279-283) and supported it by data in Table 8.

• All results of the Student's t-test, χ² test, and Pearson correlation analysis should be given in Tables or Figures. Also, the result of Kolmogorov-Smirnov and Pearson-Mises criteria should be described.

Response: In accordance with the remark, we have given all the results of the Student's test, the χ² test, Pearson's correlation analysis and the Kolmogorov-Smirnov test in tables and in the text of the article. The Pearson-Mises test was removed from the description as duplicating the results of checking the distribution normality using the Kolmogorov-Smirnov test.

• Abbreviated variables in all the Tables should be spelled out in the footnote.

Response: All abbreviated variables are explained in notes to tables and figure.

• Page 16, Lines 286-288: “In our study, the absence of correlations between the smoking index and the QoL in the rural residents suggested their riskier behavior in relation to their own health.” This reviewer is unable to understand the above statement.

Response: This statement has been rephrased in a more correct form (Page 23, Lines 354-358)

• Page 16, Lines 294-296: “One of the explanations may lie in the varying impact of environmental factors on rural and urban residents, domestic animal ownership, as well as lower intake of controller medications.” This statement is not clear to me.

Response: We have replaced this sentence with a more likely explanation (Page 24, Lines 360-362)

• The conclusions however let it down as this is more of a summary than a conclusion. Please write an appropriate conclusion, which is supported by the important outcomes of the study.

Response: We agreed with the remark and changed the conclusions.

Once again, we sincerely thank you for your thorough review and comments on the substance of the article.

Sincerely,

Dr. Natalia M. Perelman

---

## [Decision Letter · Decision Letter 1]

16 Nov 2022

PONE-D-22-00706R1Differences in the health-related quality of life in patients with asthma living in urban and rural areas in the Amur Region of Russian FederationPLOS ONE

Dear Dr. Perelman,

The reviewers have reviewed your manuscript, and have further comments for your kind response.

We look forward to receiving your revised manuscript.

Kind regards,

Munn-Sann Lye, MBBS, MPH, DrPH

Academic Editor

PLOS ONE

Reviewers' comments:

Reviewer's Responses to Questions

**Comments to the Author**

1. If the authors have adequately addressed your comments raised in a previous round of review and you feel that this manuscript is now acceptable for publication, you may indicate that here to bypass the “Comments to the Author” section, enter your conflict of interest statement in the “Confidential to Editor” section, and submit your "Accept" recommendation.

Reviewer #1: All comments have been addressed

Reviewer #2: (No Response)

2. Is the manuscript technically sound, and do the data support the conclusions?

Reviewer #1: Partly

Reviewer #2: Yes

3. Has the statistical analysis been performed appropriately and rigorously? 

Reviewer #1: I Don't Know

Reviewer #2: Yes

4. Have the authors made all data underlying the findings in their manuscript fully available?

Reviewer #1: No

Reviewer #2: (No Response)

5. Is the manuscript presented in an intelligible fashion and written in standard English?

Reviewer #1: No

Reviewer #2: Yes

6. Review Comments to the Author

Reviewer #1: Authors studied several characteristics of patients with asthma. Why only QoL is mentioned in the title?

Presented study limitations are not fully adequate. It was not a real epidemiologic study and can’t represent the entire region. Data was collected from a single city hospital and included only those patients who were treated in that hospital. Lower number of included rural patients may influence statistical analysis (in particular correlation coefficients in rural patients).

Authors should make the language more clear for readers.

The text should be shortened. All unrelated information should be deleted. Authors should present all differences and discuss it in Discussion section of the manuscript.

Abstract

"In the urban group, the correlation analysis demonstrated a clear influence of RE on patients’ own assessment of their physical status of QoL” – What is“physical status of QoL”? How scored?

Why correlation coefficients presented only in some cases? Correlation coefficients under 0.4 correspond to weak correlation.

It is difficult to understand the following statement: “The disappearance of interrelationships of some QoL domains in rural residents can be regarded as a separation of the physical, psychological and social aspects of life, which negatively affects the QoL”. It is also unclear why this statement was placed in Results section of the abstract.

Significant difference was reported only for a single SF-36 subscale “Role emotional”. Data presented in conclusion of the abstract is absent in abstract Results.

Text

It is difficult to understand authors’ idea: “Subjective perception of health and lifestyle by the members of urban and rural communities are risk factors for the development of exacerbations and progression of asthma”

“…reasons for the existing differences in the course of asthma in urban and rural residents” – Why authors were sure that this difference exists?

“…disappearance of the correlation” – Absence of correlation?

Authors stated that “the burden of disease had a more significant impact on the QoL of rural residents.” However, presented results showed no difference in asthma-specific HRQoL instrument and difference in a single domain of generic SF-36 that may not be related to asthma.

What is the base for the following statement: “rural patients with asthma are not able to comprehensively assess their current health and the risks for its deterioration. On the contrary, a relatively high level medical awareness and readiness for health preservation results in a better QoL of urban residents with asthma.”?

Reviewer #2: I have gone through the revised manuscript. The authors have complied with most of the suggestions offered by the reviewers. However, there are some minor issues:

Page 6, Lines 116-118 and 119-121: “Dyspnoea was assessed ……. more severe dyspnoea” and “The degree of dyspnea ….. more severe dyspnea.” What is the difference between these two sentences?

Tables 2-5: As suggested by the reviewers, the authors did not provide the t-value and p-value in a separate column in the tables.

Table 4: The authors did not revise the column title.

Page 15, Lines 267-268: “correlation analysis showed a negative effect of smoking (SI) on GH, PF and RP.” The correlation analysis does not imply causation (cause and effect), it only shows an association between/among the variables. Therefore, rephrase the sentence.

Tables 6-8: The authors should write the values of correlation coefficients and p-values instead of writing ‘NS,’ although the relationship is not significant.

7. PLOS authors have the option to publish the peer review history of their article (what does this mean?). If published, this will include your full peer review and any attached files.

Reviewer #1: No

Reviewer #2: No

---

## [Author Response · Author response to Decision Letter 1]

25 Feb 2023

Dear Reviewer #1,

We thank you for your great work in reviewing our manuscript and for your comments. Below are the responses to the comments and information about the corrections made:

• Authors studied several characteristics of patients with asthma. Why only QoL is mentioned in the title?

Response: We aimed to study the differences in the QoL of patients with asthma, depending on the living conditions. The remaining characteristics are used to reveal the causes and relationships of these differences. Therefore, the title reflects only the QoL.

• Presented study limitations are not fully adequate. It was not a real epidemiologic study and can’t represent the entire region. Data was collected from a single city hospital and included only those patients who were treated in that hospital. 

Response: The authors did not claim this study as epidemiological. Based on the recommendations previously put forward by the reviewers, we have already made a clarification on the location of the study. The title and text of the manuscript reflects the specific area (Amur Region) in which the research was carried out. Patients living in different villages and cities of the Amur Region come to our research clinic for a consultation, so the population of the region is widely represented in the sample. However, the limitation mentioned by the reviewer is already included in the text (Page 25, Lines 374-375).

• Lower number of included rural patients may influence statistical analysis (in particular correlation coefficients in rural patients).

Response: The difference in the number of rural and urban residents included in the study indeed may influence the absolute values of the correlation coefficients. However, it is taken into account when calculating p according to the criterion t. The absence of most relationships between QoL indicators in the group of rural respondents was confirmed statistically. 

• Authors should make the language more clear for readers.

Response: We have tried to present the results of our research as clearly as possible. Claims to the style of manuscript are most often subjective, and it is difficult for us to understand what exactly does not suit the reviewer.

• The text should be shortened. All unrelated information should be deleted. Authors should present all differences and discuss it in Discussion section of the manuscript.

Response: The text has been shortened where possible without compromising completeness. Previously, we significantly increased the volume of the text, especially in the Methods section, at the request of the reviewers themselves. Unrelated information has been removed. All differences are presented in the Results section and discussed in the Discussion section.

• Abstract. "In the urban group, the correlation analysis demonstrated a clear influence of RE on patients’ own assessment of their physical status of QoL” – What is“physical status of QoL”? How scored?

Response: We replaced the term “physical status” with the more adequate term “physical functioning” and supplemented it with a correlation coefficient between RE and PF (Page 2, Line 38). QoL domains reflect patients' perceptions of the physical, emotional, and social aspects of life. The physical health measure by SF-36 includes physical functioning (PF, ten items), role-physical (RP, four items), bodily pain (BP, two items). 

• Why correlation coefficients presented only in some cases? Correlation coefficients under 0.4 correspond to weak correlation.

Response: We have added correlation coefficient to the Results in the Abstract (Page 2, Line 38). Unfortunately, the abstract is limited by the requirements of the journal, so the following text has been changed without values of the correlation coefficients. The significance of the correlation coefficient is determined not by the absolute value, which depends on the number of correlation pairs, but by the probability the null hypothesis, which is characterized by the p value. We do not state anywhere that correlation coefficients below 0.4 correspond to a strong correlation.

• It is difficult to understand the following statement: “The disappearance of interrelationships of some QoL domains in rural residents can be regarded as a separation of the physical, psychological and social aspects of life, which negatively affects the QoL”. It is also unclear why this statement was placed in Results section of the abstract.

Response: We have changed the description of the Results section in the abstract. But we would like to give an explanation in response to the remark. In a cohort of urban residents, we revealed versatile correlations between different domains of QoL. For example, significant correlation coefficient was between PF and MH. The first indicator reflects the physical functioning, the second - the mental health, fnd both correlate with SF, which reflects social functioning. In the cohort of rural residents, there were no statistically significant correlations between SF, PF and MH. We suggest that the absence of interrelationships in some areas of QoL among rural residents can be considered as a dissociation of patients' perception of the physical, psychological and social aspects of life.

• Significant difference was reported only for a single SF-36 subscale “Role emotional”. Data presented in conclusion of the abstract is absent in abstract Results.

Response: We added Results by data on more frequent use of short-acting beta-agonists and higher incidence of clinical level anxiety in rural residents (Page 3, Lines 45-50). Thus, the data presented in the conclusion of the theses began to correspond to the results.

• It is difficult to understand authors’ idea: “Subjective perception of health and lifestyle by the members of urban and rural communities are risk factors for the development of exacerbations and progression of asthma”.

Response: We reformulated more correctly (Page 4, Lines 70-71).

• “…reasons for the existing differences in the course of asthma in urban and rural residents” – Why authors were sure that this difference exists?

Response: We present in Table 2 data on the greater need for beta-2-agonists in rural patients with asthma. In addition, references are made in the previous reformulated sentence to available literature data. However, here we have corrected to: … existing possible differences … (Page 4, Line 87)

• “…disappearance of the correlation” – Absence of correlation?

Response: We corrected to “…absence of the correlation…” (Page 14, Lines 262-263).

• Authors stated that “the burden of disease had a more significant impact on the QoL of rural residents.” However, presented results showed no difference in asthma-specific HRQoL instrument and difference in a single domain of generic SF-36 that may not be related to asthma.

Response: We agree that the differences we have identified from the questionnaire data may not be convincing enough for this statement. Taking into account the data of the correlation analysis, which showed the absence of many correlations between various QoL domains in rural residents compared to urban residents, we changed the first sentence in the Conclusions: “The burden of asthma introduces a greater imbalance in the health-related QoL of rural residents compared to urban residents in the Amur region of the Russian Federation” (Page 25, Lines 382-384).

• What is the base for the following statement: “rural patients with asthma are not able to comprehensively assess their current health and the risks for its deterioration. On the contrary, a relatively high level medical awareness and readiness for health preservation results in a better QoL of urban residents with asthma.”?

Response: This statement is based on evidence of a lower level of education in the cohort of rural residents in the study (Page 5, Lines 101-104). In addition, the lower level of education, culture and availability of qualified medical care among rural residents in the Russian Federation compared to urban residents is reflected in the source [20]. Similar data are given in [16]: “Overall rural US populations face increased poverty and inferior health care for reasons related to insurance status and to poor access due to increased travel distance and lack of providers, particularly specialists”. We have changed the sentence to express this statement in a less categorical form: “Rural patients with asthma are more often unable comprehensively assess…” (Page 23, Lines 326-327).

Once again, thank you for your review and comments.

Sincerely,

Dr. Natalia M. Perelman 

Dear Reviewer #2,

We thank you for your work in reviewing our manuscript and for your comments. Below are the responses to the comments and information about the corrections made:

• Abstract: Page 6, Lines 116-118 and 119-121: “Dyspnoea was assessed ……. more severe dyspnoea” and “The degree of dyspnea ….. more severe dyspnea.” What is the difference between these two sentences?

Response: We removed the erroneous duplicate.

• Tables 2-5: As suggested by the reviewers, the authors did not provide the t-value and p-value in a separate column in the tables.

Response: We added to the table 2-5 columns with t values.

• Table 4: The authors did not revise the column title.

Response: We have corrected the column title in Table 4.

• Page 15, Lines 267-268: “correlation analysis showed a negative effect of smoking (SI) on GH, PF and RP.” The correlation analysis does not imply causation (cause and effect), it only shows an association between/among the variables. Therefore, rephrase the sentence.

Response: We have rephrased the sentence (Page 15, Lines 271-272).

• Tables 6-8: The authors should write the values of correlation coefficients and p-values instead of writing ‘NS,’ although the relationship is not significant.

Response: We have written the correlation coefficients and p values in Table 6-8.

Once again, we sincerely thank you for your review and comments.

Sincerely,

Dr. Natalia M. Perelman

---

## [Decision Letter · Decision Letter 2]

20 Mar 2023

PONE-D-22-00706R2Differences in the health-related quality of life in patients with asthma living in urban and rural areas in the Amur Region of Russian FederationPLOS ONE Dear Dr. Perelman,

Thank you for submitting your manuscript to PLOS ONE. After careful consideration, we feel that it has merit but does not fully meet PLOS ONE’s publication criteria as it currently stands. Therefore, we invite you to submit a revised version of the manuscript that addresses the points raised during the review process.

Thank you for your responses to the reviewers who have found your manuscript acceptable.

However, on further examination of the manuscript, I have noticed some discrepancies as follows:

1. Lines 363-365: "Perhaps this is due to the reported higher frequency of use of short-acting β2-agonists by urban residents."

This contradicts the results contained in Lines 201-202 - "we noted a significantly higher need for β2-agonists in rural areas compared to urban areas." as well as the data on beta-2 agonists in Table 2

where urban patients had 2.7 ± 0.3 inh/day vs rural patients with 4.2 ± 0.6 inh/day.

In addition, this sentence (Lines 363-365) does not follow logically from the previous sentence in the context of dyspnea among urban patients.

Perhaps you mean "lower" frequency in Lines 363-365?

2. Lines 257-258: "The general health (GH) positively correlated with most of the SF-36 scales except PF and VT." This sentence refers to urban patients and therefore data contained in table 6.

In table 6, the correlation coefficient for the correlation between GH and VT is r=0.19, p=0.0085, which is significant and therefore contradicts the statement above regarding VT.

Some other minor corrections to be made:

Line 49: should read "45 persons" instead of "45 person".

Line 242: should read "45 persons" instead of "45 person".

Lines 326-327: should read "unable to comprehensively assess" instead of "unable comprehensively assess".

Line 332: should read "absence" instead of "disappearance".

Kindly have the above discrepancies and errors addressed and corrected.

We look forward to receiving your revised manuscript.

Kind regards,

Munn-Sann Lye, MBBS, MPH, DrPH

Academic Editor

PLOS ONE

Journal Requirements:

Reviewers' comments:

Reviewer's Responses to Questions

**Comments to the Author**

1. If the authors have adequately addressed your comments raised in a previous round of review and you feel that this manuscript is now acceptable for publication, you may indicate that here to bypass the “Comments to the Author” section, enter your conflict of interest statement in the “Confidential to Editor” section, and submit your "Accept" recommendation.

Reviewer #1: All comments have been addressed

Reviewer #2: (No Response)

2. Is the manuscript technically sound, and do the data support the conclusions?

Reviewer #1: Partly

Reviewer #2: Yes

3. Has the statistical analysis been performed appropriately and rigorously? 

Reviewer #1: Yes

Reviewer #2: Yes

4. Have the authors made all data underlying the findings in their manuscript fully available?

Reviewer #1: No

Reviewer #2: Yes

5. Is the manuscript presented in an intelligible fashion and written in standard English?

Reviewer #1: Yes

Reviewer #2: Yes

6. Review Comments to the Author

Reviewer #1: (No Response)

Reviewer #2: (No Response)

7. PLOS authors have the option to publish the peer review history of their article (what does this mean?). If published, this will include your full peer review and any attached files.

Reviewer #1: No

Reviewer #2: No

---

## [Author Response · Author response to Decision Letter 2]

22 Mar 2023

Dear Editor,

Thanks for the necessary clarification. We provide below the answer to your last comment:

1. In the Methods section please include the informed consent statement to reflect whether "written or verbal" informed consent was obtained from all participants for inclusion in the study.

Response: We supplemented the informed consent statement with information about signing a written informed consent (Page 5, Lines 108-109): “All patients, after preliminary familiarization with the study protocol, signed written informed consent.”

---

## [Editor Report · Decision Letter 3]

6 Apr 2023

Differences in the health-related quality of life in patients with asthma living in urban and rural areas in the Amur Region of Russian Federation

PONE-D-22-00706R3

Dear Dr. Perelman,

We’re pleased to inform you that your manuscript has been judged scientifically suitable for publication and will be formally accepted for publication once it meets all outstanding technical requirements.

Kind regards,

Munn-Sann Lye, MBBS, MPH, DrPH

Academic Editor

PLOS ONE
---

## [Editor Report · Acceptance letter]

29 Jun 2023

PONE-D-22-00706R3 

Differences in the health-related quality of life in patients with asthma living in urban and rural areas in the Amur Region of Russian Federation 

Dear Dr. Perelman:

I'm pleased to inform you that your manuscript has been deemed suitable for publication in PLOS ONE. Congratulations! Your manuscript is now with our production department. 

Kind regards, 

on behalf of

Professor Munn-Sann Lye 

%CORR_ED_EDITOR_ROLE%

PLOS ONE